# High-Frequency Transcranial Random Noise Stimulation Modulates Gamma-Band EEG Source-Based Large-Scale Functional Network Connectivity in Patients with Schizophrenia: A Randomized, Double-Blind, Sham-Controlled Clinical Trial

**DOI:** 10.3390/jpm12101617

**Published:** 2022-09-30

**Authors:** Ta-Chuan Yeh, Cathy Chia-Yu Huang, Yong-An Chung, Jooyeon Jamie Im, Yen-Yue Lin, Chin-Chao Ma, Nian-Sheng Tzeng, Hsin-An Chang

**Affiliations:** 1Department of Psychiatry, Tri-Service General Hospital, National Defense Medical Center, Taipei 114201, Taiwan; 2Department of Life Sciences, National Central University, Taoyuan 320317, Taiwan; 3Department of Nuclear Medicine, College of Medicine, The Catholic University of Korea, Seoul 21431, Korea; 4Department of Emergency Medicine, Tri-Service General Hospital, National Defense Medical Center, Taipei 114202, Taiwan; 5Department of Emergency Medicine, Taoyuan Armed Forces General Hospital, Taoyuan 325208, Taiwan; 6Department of Psychiatry, Tri-Service General Hospital Beitou Branch, National Defense Medical Center, Taipei 112003, Taiwan

**Keywords:** transcranial random noise stimulation, electroencephalography, functional connectivity, schizophrenia, negative symptoms

## Abstract

Schizophrenia is associated with increased resting-state large-scale functional network connectivity in the gamma frequency. High-frequency transcranial random noise stimulation (hf-tRNS) modulates gamma-band endogenous neural oscillations in healthy individuals through the application of low-amplitude electrical noises. Yet, it is unclear if hf-tRNS can modulate gamma-band functional connectivity in patients with schizophrenia. We performed a randomized, double-blind, sham-controlled clinical trial to contrast hf-tRNS (N = 17) and sham stimulation (N = 18) for treating negative symptoms in 35 schizophrenia patients. Short continuous currents without neuromodulatory effects were applied in the sham group to mimic real-stimulation sensations. We used electroencephalography to investigate if a five-day, twice-daily hf-tRNS protocol modulates gamma-band (33–45 Hz) functional network connectivity in schizophrenia. Exact low resolution electromagnetic tomography (eLORETA) was used to compute intra-cortical activity from regions within the default mode network (DMN) and fronto-parietal network (FPN), and functional connectivity was computed using lagged phase synchronization. We found that hf-tRNS reduced gamma-band within-DMN and within-FPN connectivity at the end of stimulation relative to sham stimulation. A trend was obtained between the change in within-FPN functional connectivity from baseline to the end of stimulation and the improvement of negative symptoms at the one-month follow-up (r = −0.49, *p* = 0.055). Together, our findings suggest that hf-tRNS has potential as a network-level approach to modulate large-scale functional network connectivity pertaining to negative symptoms of schizophrenia.

## 1. Introduction

Schizophrenia is characterized by positive symptoms, negative symptoms, and cognitive impairments. The domain of negative symptoms (e.g., alogia, anhedonia, amotivation, asociality, and affective flattening) has enduring and substantial functional impact on the patients [1]. Moreover, current antipsychotics have little, if any, clinically-relevant impact on primary negative symptoms [2]. Addressing the unmet need will require a new therapeutic approach. Recent research indicates transcranial electrical stimulation (tES) is a representative non-invasive brain stimulation method (NIBS) that may be effective in ameliorating negative symptoms [3]. Typical types of tES methods include transcranial direct current stimulation (tDCS), transcranial alternating current stimulation (tACS), and transcranial random noise stimulation (tRNS). In our recent pilot work, adjunct high-frequency transcranial random noise stimulation (hf-tRNS) over the lateral prefrontal cortex has been shown to rapidly improve negative symptoms in a cohort of stabilized schizophrenia patients [4]. tRNS is a form of subthreshold electrical stimulation of cortical neurons, inducing neural noise by delivering a low-intensity alternating current in random amplitudes and frequencies [5]. Research indicates that tRNS may be the most effective transcranial electrical stimulation (tES) method to increase cortical excitability and thereby elicit behavioral changes [6]. The high-frequency range (101–640 Hz) of tRNS has been reported to increase neuronal excitability of the stimulated cortex possibly through the repetitive opening of the voltage-gated Na^+^ channels [5] and modulate event-related gamma-band oscillatory activity [7] and hemodynamic response of functional stimulus [8] possibly via the stochastic resonance (SR) that enhances neural signal-to-noise ratio and neuronal synchronization within the stimulated cortex. However, the neural working mechanisms underlying the prompt attenuation of negative symptoms by hf-tRNS over the lateral prefrontal cortex remain to be determined

Synchronous neural oscillations represent an underlying mechanism responsible for connectivity or communication within multiple large-scale brain networks. The dysconnection hypothesis posits that schizophrenia is associated with altered functional connectivity (i.e., either hypoconnectivity or hyperconnectivity between distinct brain regions). Recent studies have linked the complex psychopathological symptoms of schizophrenia to malfunctioning of two higher-order functional networks: the default-mode network (DMN) and fronto-parietal network (FPN) [9]. The DMN is thought to be engaged in self-relevant internal information processing with key nodes in the medial prefrontal cortex (MPFC), posterior cingulate cortex (PCC), precuneus cortex, and bilateral angular gyri (AG). The FPN, primarily composed of the dorsolateral prefrontal cortex (DLPFC) and posterior parietal cortex, is involved in multiple executive functions. Increasing evidence suggests that functional integration and segregation these large-scale neuronal networks are critical for self-experience and mental health [10]. Specifically, aberrant functional interactions within these brain networks may have particular relevance for self-experience alterations (e.g., psychosis-like experiences) and an increased risk for psychosis in healthy individuals [11,12]. Neuroimaging studies in schizophrenia have depicted that the aberrant connections within and between DMN and FPN as revealed by resting-state functional magnetic resonance imaging (fMRI) are correlated with negative symptom burden and negative symptom reductions following treatment [9,13]. Compared to fMRI, electroencephalography (EEG) can directly reflect brain electrical activity and can be further analyzed into multiple frequency bands, allowing for diverse interpretations according to the characteristics of each frequency band. There have been some resting-state EEG source-based studies in schizophrenia patients showing abnormally organized brain network (e.g., DMN) connectivity for the different EEG frequency components [14,15].

Among the different EEG frequency components, gamma-band oscillations are of growing interest due to their involvement in neural synchronization of both local and large-scale brain networks underlying higher-order perceptual and cognitive processing that was impaired in schizophrenia [16]. The generation of gamma-band oscillations relies on a closed feedback loop involving glutamatergic pyramidal cells and parvalbumin-positive GABAergic interneurons. The glutamate N-methyl-D-aspartate receptors (NMDAR) on these GABAergic interneurons are fundamental for the synchronized inhibition to generate gamma-band oscillations. A glutamate NMDAR dysfunction has been suggested as a possible mechanism underlying altered connectivity in schizophrenia. Research points to increased resting-state gamma-band activity and connectivity in patients with schizophrenia and reveals a link between abnormally organized brain networks in the gamma band and the core symptoms of schizophrenia [17,18,19]. More specifically, research has implicated a disruption of GABAergic-glutamatergic balance in the etiology of negative symptoms [20] and suggests that resting-state gamma-band connectivity could be a potential biomarker that can help predict or monitor the response of negative symptoms to novel treatment [21].

Motivated by the aforementioned findings, this study aimed to investigate if applying 10 sessions of 20 min hf-tRNS to the lateral prefrontal cortex over 5 days modulates the gamma-band between- and within-network functional connections of DMN and FPN in patients with schizophrenia and whether the characteristics of network connectivity serve as predictors or surrogates of treatment response for personalized intervention. Clinical data for this study have been reported elsewhere [4]. We did not construct specific hypotheses regarding the direction of changes in DMN and FPN gamma-band network connectivity due to the paucity of relevant previous studies.

## 2. Materials and Methods

### 2.1. Participants

This study was performed at the Tri-Service General Hospital (TSGH) in Taipei, Taiwan (ClinicalTrials.gov accessed on 31 July 2019, NCT04038788) and was approved by the Institutional Review Board of TSGH. The trial recruited participants who were diagnosed with schizophrenia or schizoaffective disorder and were symptomatically stable on antipsychotic treatments from a single academic medical center. The diagnosis was confirmed by a psychiatrist (H.C.) using diagnostic interview based on the DSM-5 criteria. Patients were included in this study if they: (1) were aged 20–65; (2) had an illness duration ≥ 1 year; (3) were stabilized on an adequate therapeutic dose of antipsychotics for at least 8 weeks before enrolment; (4) had a Positive and Negative Syndrome Scale (PANSS) total score < 120 at both screening and baseline; and (5) agreed to participate in the study and provided written informed consent. Participants were excluded from the study if they: (1) had any active substance use disorder (exception for caffeine and/or tobacco) or current psychiatric comorbidity; (2) had implanted metal or cerebral medical devices in the head; (3) were pregnant or lactating women; and (4) had any history of cerebrovascular diseases, seizures, intracranial neoplasms/surgery, or severe head injuries. Table 1 showed the concise demographics and clinical assessments of the participants. The study analyzed data from 35 participants. There were no significant differences in the sociodemographic and clinical characteristics between hf-tRNS group and sham group.

### 2.2. Study Design

This study performed between October 2018 and May 2019 was a randomized, double-blind, and sham-controlled clinical trial, with two stimulation conditions (hf-tRNS and sham). Each participant was randomly assigned to one of the two conditions (i.e., hf-tRNS:sham = 1:1). A study coordinator not involved in the trial created 5-digit random numbers and assigned the randomization numbers to the participants to accomplish the blinding integrity (Appendix A). The participants and researchers were unaware of the group assignments until unblinding of the trial. The negative symptom severity as the primary outcome was measured by the Positive and Negative Syndrome Scale Factor Score for Negative Symptoms (PANSS-FSNS), i.e., the PANSS items N1-4, N6, G7, and G16 [22]. The PANSS-FSNS was rated at baseline, at the end of stimulation, and the one-week and one-month follow-up visits. Detailed analyses of clinical data have been reported in a separate paper [4].

### 2.3. The Sample Size Calculation of the Participants

G * power 3.1.9.4 was used to calculate the sample size for repeated measure analysis of variance “treatment group” × “time” interaction effect for primary outcome measurement with effect size = 0.25, alpha error probability = 0.05, power = 0.95 (two tailed), number of groups = 2, number of measurements = 4, and correlation among repeated measures = 0.5. The calculated total sample size was 36.

### 2.4. Brain Stimulation

A battery-operated device (Eldith Stimulator Plus, NeuroConn, Ilmenau, Germany) connected with an equalizer extension box for the high-definition 4 × 1 electrode montage was used for stimulation. Figure 1 and Figure 2 illustrated the electrode layout and modeling of electric field distribution, respectively. Five carbon rubber electrodes of radius 1 cm with anode placed over AF3 of the international 10-10 EEG system and cathodes over AF4, F2, F6, and FC4 (Figure 1) were applied to the scalp with Ten20 conductive paste (Bio-Medical Instruments, Clinton Township, Michigan, MI, USA) and were checked for the combined impedance of all electrodes < 15 kΩ. Active hf-tRNS delivered twice-daily, 20 min, 2 mA-intensity random noise stimulation with 100–640 Hz frequency, 1 mA offset, and 15 s ramp-in/ramp-out for 5 consecutive weekdays. Sham stimulation delivered 40 s, 2 mA normal-like stimulation, followed by a tiny current pulse (110 μA over 15 ms) for impedance control taking place every 550 ms for the remaining time. The participants sat comfortably and kept their eyes open during stimulation unless otherwise specified. The break between twice-daily stimulation sessions was at least 2 h.

### 2.5. Randomization and Blinding

A study coordinator not involved in the execution of the clinical trial performed the randomization by using a web-based randomization tool (https://www.sealedenvelope.com/) accessed on 24 October 2018. Participants were assigned to the active or sham stimulation in a 1:1 ratio using blocked randomization with randomly permuted blocks of four. The allocation concealment system was performed through central randomization, in which the researcher contacted the study coordinator after enrolling and registering the participant. The allocation concealment was further ensured by the administration of hf-tRNS using “study mode of the device” in which a five-digit numerical code specific to individual participant was entered into the device (Eldith DC stimulator Plus, NeuroConn, Ilmenau, Germany) that resulted in either active or sham stimulation (i.e., the researcher got the randomization code and a unique five-digit numerical code for an individual participant from the study coordinator while hf-tRNS administrator entered the code for study mode into the device). The study coordinator had continuous access to the randomization list and unblinded the study after the final visit of the last participant. Not until the unblinding of the trial did the participants, hf-tRNS administrators, researchers and clinical raters know the actual stimulation types. There was only one reason for premature code-breaking and that was when any suspected unexpected serious adverse reaction (SUSAR) occurred. The treatment code was be broken by the study coordinator before reporting a SUSAR to the health agency and the local institutional review boards (IRB). The treatment for the participant would be discontinued when his/her masking code was broken

### 2.6. EEG at Rest

Resting-state EEG (rsEEG) was collected at baseline, at the end of stimulation and the one-week follow-up, using a 32-channel EEG cap (NP32, GmbH, Ilmenau, Germany) with Ag/AgCl sintered ring electrodes placed according to the international 10–20 system and referenced to the tip of the nose, together with Neuro Prax^®^ TMS/tES compatible full band DC-EEG system (NeuroConn GmbH, Ilmenau, Germany) with a sampling frequency of 4000 Hz, an analogue-digital precision of 24 bits, and an analogous bandpass filter (0–1200 Hz). Patients were seated comfortably in a recliner in a light and sound attenuated room. They were instructed not to drink caffeinated beverages one hour before EEG recording and alcohol 24 h before recording to avoid caffeine- or alcohol-induced changes in the EEG stream. The ground electrode was placed at Fpz. Horizontal electrooculogram (HEOG) was recorded by two electrodes placed at 1 cm from the outer canthi of both eyes. Two electrodes were placed above and below the left eye, respectively, to record the blinks and vertical electrooculogram (VEOG). The impedance of each electrode was checked to remain below 5 kΩ. Before starting the EEG recording, the patients performed 3-min calibration tasks to estimate the influence of horizontal/vertical movements and blink artifacts on EEG, which was processed and stored in the Neuro Prax^®^ EEG system built-in software providing fully automatic correction of real-time EEG artifacts caused by blinking and eye or body movement during the subsequent EEG recording (Appendix A). rsEEG with eyes open (5 min) and eyes closed (5 min) were recorded for a total of 10 min and the sequence was randomized and counterbalanced across the patients. Patients were instructed to visually fixate on a crosshair in front of them during the eyes-open condition or stay relaxed in a state of mind wandering (i.e., without goal-oriented mental activity) with their eyes closed during the eyes-close condition. Offline, the data were downsampled to 500 Hz, band-pass filtered to 1–100 Hz with the Finite Impulse Response (FIR) method (i.e., using the tool of Basic FIR filter (new, default) with 1 Hz as the lower edge frequency and 100 Hz as the higher edge frequency) and analog 60 Hz-notch filtered using EEGLAB v2020.0 [23], an interactive Matlab (MathWorks, Natick, MA, USA) toolbox. Bad channels were automatically detected and removed based on artifact subspace reconstruction (ASR) [24]. Independent component analysis (ICA) followed by ICLabel [25] was used to automatically remove artifacts caused by muscle activity, heartbeats, eye movements, and eye blinks. Since hf-tRNS was applied in an eyes-open state, only accepted epochs of eyes-open EEG data collected in a resting state were selected for power spectral analysis using fast Fourier transforms to obtain spectral estimates of gamma (33–45 Hz) oscillations.

### 2.7. Electrical Source Estimation

All source imaging was performed with the exact low resolution brain electromagnetic tomography (eLORETA) software that computes the exact magnitude of cortical activity as current density (A/m^2^) by correctly localizing and reconstructing the intracerebral electrical sources underlying the scalp-recorded activity [26] in a realistic head model [27], using the Montreal Neurological Institute (MNI; Montreal, Quebec, QC, Canada) MNI152 template [28], with the three-dimensional eLORETA inverse solution space (i.e., intracerebral volume) restricted to cortical gray matter and hippocampi, as determined by the probabilistic Talairach atlas [29], and partitioned in 6239 voxels (voxel size 5 mm× 5 mm× 5 mm). The eLORETA images matching the estimated neuronal generators of brain activity within the gamma frequency band were calculated.

### 2.8. Source-Based Functional Connectivity

EEG source-based functional connectivity was computed by using the validated eLORETA algorithm [30]. Lagged phase synchronization (LPS) is a measure that estimates the phase synchronization between two signals in the frequency domain based on normalized Fourier transforms after excluding the zero-lag, instantaneous component of phase synchronization caused by intrinsic artifacts or non-physiological effects [26]. The LPS between two brain regions is thought to truly represent the interregional physiological connectivity. Seeds from key regions within the DMN and FPN selected from the seven-network parcellation [31] were predetermined to create regions of interest (ROIs) in eLORETA (see MNI coordinates of these chosen seeds in Table 2). Subcortical seeds were omitted, and bilateral seeds close to the midline were fused into a single seed. ROIs (10 from the DMN and 9 from the FPN) were defined by encompassing all gray matter voxels within a 15 mm radius of the seed points. The functional connectivity within- and between-network were examined by simultaneously computing LPS between any pair of ROIs in the DMN and FPN for artifact-free EEG segments in gamma frequency band.

### 2.9. Statistical Analyses

Statistical analyses were performed either using IBM SPSS Statistics 21.0 software (IBM SPSS Inc., Chicago, IL, USA) or the implemented statistical eLORETA nonparametric mapping (SnPM) tool [26]. The SnPM analysis tool includes a correction for multiple comparisons. The effects of hf-tRNS on gamma-band power at the scalp level over time were analyzed using repeated measures analysis of variance (RMANOVA) including “time” as the within-group factor (baseline, the end of stimulation, and the one-week follow-up) and “treatment group” (active versus sham) as the between-group factor. Post-hoc statistical tests were performed using student’s *t*-test and multiple comparisons were corrected by the false discovery rate (FDR) method. The statistical analyses of between-group changes in gamma-band electrical source estimation and source functional connectivity from baseline to post-baseline visits were conducted using *t*-tests that were corrected for multiple comparisons using a non-parametric permutation procedure (5000 randomizations). Spearman rank correlations were used to analyze the relationships between the changes in EEG-based measures from baseline to post-baseline visits and treatment response to hf-tRNS. Statistical significance for the results was set at *p* < 0.05 (two-tailed) and the FDR was used for multiple comparisons correction.

## 3. Results

### 3.1. Effects of hf-tRNS on Scalp- and Source-Level Gamma-Band Power

At individual electrode level, there were no significant differences in absolute gamma-band power at baseline between hf-tRNS group and sham group. RMANOVA did not show any significant group-by-time interaction for absolute gamma-band power at individual electrode level (all *p* values > 0.05). eLORETA was applied to localize the changes in gamma activity from baseline to each post-baseline visit. No significant differences in gamma-band current densities (eLORETA) at baseline were found between hf-tRNS group and sham group. SnPM did not show any significant changes in gamma-band current densities from baseline to each post-baseline visit in hf-tRNS condition compared to sham (all *p* values > 0.05).

### 3.2. Effects of hf-tRNS on Source-Based Gamma-Band within-Network Connectivity

SnPM showed significant between-group changes in gamma-band within-DMN connectivity from baseline to the end of stimulation (Figure 3). The LPS was significantly reduced in hf-tRNS group compared with sham, between a region in the medial frontal gyrus (MFG) and the regions in the left angular gyrus (AG), left parahippocampal gyrus (PHG), and posterior cingulate (PC), and between a region in the right superior frontal gyrus (SFG) and a region in the left SFG (all *p* values < 0.05, corrected). SnPM also showed significant between-group changes in within-FPN connectivity from baseline to the end of stimulation. The LPS was significantly reduced in hf-tRNS group compared with sham, between a region in the right frontal pole (FP) and a region in the left middle temporal gyrus (MTG), and between a region in the left supramarginal gyrus (SMG) and a region in the cingulate gyrus (CG) (all *p* values < 0.05, corrected). The inclusion of antipsychotic medication dose (in chlorpromazine equivalents) did not substantially alter the results. However, there were no significant between-group changes in gamma-band within-DMN and within-FPN connectivity from baseline to the one-week follow-up.

### 3.3. Effects of hf-tRNS on Source-Based Gamma-Band between-Network Connectivity

The changes in gamma-band between-network (DMN-FPN) connectivity from baseline to any post-baseline visits were not significantly different between hf-tRNS group and sham (all *p* values > 0.05).

### 3.4. Effects of hf-tRNS on Source-Based Gamma-Band Whole-Brain Functional Connectivity

In addition to eLORETA seed-based analyses with a-priori selected seeds for large-scale network functional connectivity analyses, the whole-brain analyses were further reported in the Appendix A to avoid biases. Appendix A and Appendix A showed that patients treated with hf-tRNS had reduced functional connectivity in several large-scale brain networks at the end of stimulation in comparison with the sham group (e.g., connectivity between right middle frontal gyrus and left posterior cingulate, between right middle frontal gyrus and left cuneus, between right lingual gyrus and bilateral anterior cingulate, between right cingulate gyrus and right paracentral lobule, and between right superior temporal gyrus and left posterior cingulate, all *t* values > 3.22, two-tailed *p* values < 0.01, Appendix A).

### 3.5. Correlation Analyses

In the hf-tRNS group, gamma-band within-DMN, within-FPN, and between-network (DMN-FPN) connectivity at baseline failed to predict treatment response at the end of stimulation and the follow-up visits when antipsychotic medication dose (in chlorpromazine equivalents) was controlled. A significant trend was obtained between the change in left SMG-CG (two regions in the FPN) functional connectivity from baseline to the end of stimulation and the improvement of negative symptoms at the one-month follow-up (*r* = −0.49, *p* = 0.055, Appendix A). However, the critical level of significance for the association was not reached after FDR correction. The results of other correlation analyses were all non-significant.

## 4. Discussion

To our knowledge, this is the first randomized, double-blind, sham-controlled clinical trial providing evidence that adjunct hf-tRNS improves negative symptoms of schizophrenia through modulating gamma-band EEG source-based large-scale functional network connectivity. Recent research showed that a single session of hf-tRNS with a low current intensity (1 mA) could modulate brain oscillatory activity within gamma band in healthy human individuals [7]. There was a lack of evidence for the acute and longer-lasting effects of repetitive hf-tRNS using a protocol comprising 2 mA, 20 min for 10 s (i.e., a total stimulation duration of 200 min) on gamma-band local neural oscillations and long-range functional connectivity in patients with schizophrenia. In the present study, hf-tRNS was applied to the prefrontal cortex for targeting EEG gamma-band oscillations of two higher-order neural networks (DMN and FPN). The results of clinical data showed that hf-tRNS treated negative symptoms effectively in stable patients with schizophrenia [4] while the EEG results showed that within-DMN (Figure 3) and within-FPN (Figure 4) functional connectivity in the gamma-frequency band were significantly reduced at the end of stimulation among participants treated with hf-tRNS relative to sham condition. Evidence indicates that patients with schizophrenia showed increased EEG-based resting-state functional connectivity at gamma frequency compared to the controls, especially within a distinct and strongly lateralized network consisting mainly of the left fronto-parieto-temporal networks (e.g., several nodes of the DMN and FPN, including left inferior frontal/orbitofrontal, lateral and medial temporal, and inferior parietal areas) [18]. However, the cross-sectional nature of a case-control study precludes a definite conclusion to the question of whether increased gamma-band long-range functional connectivity reflects a successful compensatory mechanism for the brain to adapt to the changes brought about by the pathophysiological process of schizophrenia or it represents a primary abnormality underlying the psychopathological symptoms (e.g., negative symptoms, which are more relevant for the present study). Our study demonstrated that the adjunct 10 sessions of hf-tRNS resulted in significantly reduced gamma-band functional connectivity within DMN and FPN (i.e., interregional de- synchronization) along with the improvement in negative symptoms, despite the fact that significant changes in scalp- and source-level gamma-band power (i.e., local phase-coupling) were not observed in our sample. These findings suggest that gamma-band long-range functional hyper-connectivity could be a neural signature of negative symptoms of schizophrenia and successful reshaping of gamma-band large-scale functional connectivity may have led to the decreased negative symptoms observed in the participants. Although grand-averaged outlasting effects of hf-tRNS on gamma-band large-scale functional connectivity at the one-week follow-up did not reach significance, some participants with more persistent clinical improvement did show outlasting effects on neural network connectivity. In other words, changes in brain network dynamics in response to hf-tRNS appear to a valuable tool that has the potential to track the therapeutic effect of this novel non-invasive brain stimulation. Taken together, our results indicate that hf-tRNS was effective in modulating gamma-band large-scale functional network connectivity in patients with schizophrenia. However, this change did not have a linear correlation with change in clinical symptoms. Our study lends support to the hypothesis that the integrative functioning of brain networks (e.g., DMN and FPN) maintains a multidimensional sense of self and mental health [10,12]. Our results are also consistent with the notion of aberrant long-range connectivity for schizophrenia that posits that an intervention with the ability to modulate brain network functional connectivity could serve to normalize dysfunction in perturbed networks and thereby improve the severity of core symptoms. There are several proposed mechanisms of action for the effects of hf-tRNS on the brain (e.g., changes in cortical excitability or cortical oscillations, stochastic resonance phenomenon, and the increased sensitivity of neuronal networks to modulation). Of particular relevance to the present study is the fact that the optimal level of external noise delivered by hf-tRNS contributes to modulating neural signal-to-noise ratio and promoting neural desynchronization. Specifically, hf-tRNS intervention in schizophrenia could be working by desynchronizing the neuronal networks whose over-synchronization accounts for the psychopathology observed (i.e., negative symptoms) [32]. The electric field simulation (Figure 2) shows that the peak electric fields induced by hf-tRNS over the prefrontal cortex involve the areas of medial prefrontal cortex (a region in the DMN) and dorsolateral prefrontal cortex (a region in the FPN). It seems plausible that hf-tRNS desynchronizes the within-DMN and within-FPN hyper-connectivity in the gamma frequency through the two hubs of DMN and FPN serving as gateways into the functional networks. Due to the inherent limitations of the eLORETA source localization, however, the impact of hf-tRNS on the connectivity of other important large-scale networks implicated in schizophrenia cannot be excluded (e.g., striatal-cortical DMN and striatal-cortical FPN connectivity, which play a critical role in the aberrant-salience hypothesis of psychosis) [11]. As such, our findings await replication and further investigation using resting-state functional MRI that has superior spatial resolution.

### Limitations

Our study had limitations. First, correlation coefficients of the reduction in left SMG-CG functional connectivity on treatment response trended towards, but did not reach, statistical significance. Although we cannot exclude the possibility of a significant non-linear correlation between the two variables not detected by spearman correlation analyses, replication in a larger sample will be a practical way to allow a more precise estimate of left SMG-CG functional connectivity as a surrogate endpoint for treatment response. Second, EEG is limited in its spatial resolution recorded using the 32-channel array of scalp electrodes for source localization [33] and in its ability to detect sources of electrical activity at deep structures (e.g., cerebellum) also implicated in altered connectivity in schizophrenia [34], even though it provides a direct measure of fast neural network dynamics with millisecond temporal resolution for studying the high temporal dynamics of the network functional connectivity. Our study mainly focused on eLORETA seed-based analyses with a-priori selected seeds for large-scale network functional connectivity analyses [19,35] rather than eLORETA whole-brain analyses. It is noteworthy that hf-tRNS reduced functional connectivity of cuneus, lingual gyrus, superior temporal gyrus and paracentral lobule in large-scale brain networks, as can be seen in the results of whole-brain analyses (Appendix A and Appendix A). Research indicates that these networks play an unneglectable role in the pathophysiology of schizophrenia [36,37]. However, a potentially limiting factor for eLORETA whole-brain analyses could be the relatively low spatial resolution allowed by the number of scalp channels used in our EEG recordings, which may jeopardize the validity of eLORETA as a linear inverse solution to the inverse problem of EEG source localization. Third, all participants were currently receiving antipsychotic medication. Although direct effects of antipsychotic medication are unlikely in the present study since chlorpromazine equivalent dose had no significant effect on study results, we cannot exclude the possibility that the changes observed in gamma-band long-range connectivity were mediated by the effects of the interaction between hf-tRNS and antipsychotic medications. Fourth, the statement and interpretation regarding the resting-state gamma activity or gamma-band large-scale network connectivity in our patients with schizophrenia should be treated with caution since this study did not have a control group of non-schizophrenia individuals. Further studies including both patients and healthy subjects are required to confirm our results. Finally, neural oscillations of other frequency ranges also implicated in the altered functional connectivity for schizophrenia were not explored in the present study. For example, aberrant oscillations in the theta frequency are of interest since they are also involved in parvalbumin-positive GABAergic interneurons and since they interact with gamma-band oscillations in a meaningful way (i.e., theta–gamma coupling). Recently, some resting-state EEG source-based studies showed theta-band DMN hyperconnectivity in patients with schizophrenia [14,15]. In light of these findings, it would be of particular relevance to further investigate the effects of hf-tRNS on long-range cross-frequency coupling between theta- and gamma-band oscillations in schizophrenia.

## 5. Conclusions

In summary, we believe that the current findings allow for a better understanding of the neural impact of hf-tRNS and have the potential to inform improvements for specifically targeting negative symptoms of schizophrenia with sub-threshold non-invasive brain stimulation. These findings suggest that modulating connectivity within higher-order functional networks in the gamma frequency by using hf-tRNS over the prefrontal cortex may play a key role in reducing negative symptoms in patients with schizophrenia. In this respect, our results open the possibility for a tailor-made use of hf-tRNS targeting negative symptoms.

## Figures and Tables

**Figure 1 jpm-12-01617-f001:**
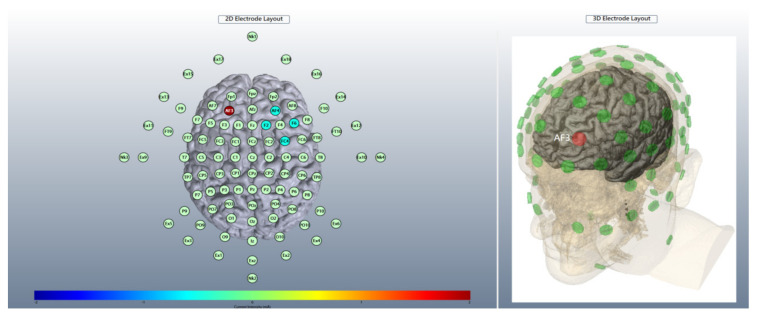
The left panel represents the 2D electrode layout of high-definition high-frequency transcranial random noise stimulation (hf-tRNS) over the lateral prefrontal cortex. According to the International 10–10 electroencephalogram electrode position, the anode (red) was placed at AF3 (current intensity: +2 mA) and the cathodes (cyan) were placed at AF4 (−0.5 mA), F2 (−0.5 mA), F6 (−0.5 mA), and FC4 (−0.5 mA). The right panel is the corresponding 3D electrode layout of the anode (red) AF3. The figure was created using HD-Explore^®^ software (Soterix Medical, New York, NY, USA).

**Figure 2 jpm-12-01617-f002:**
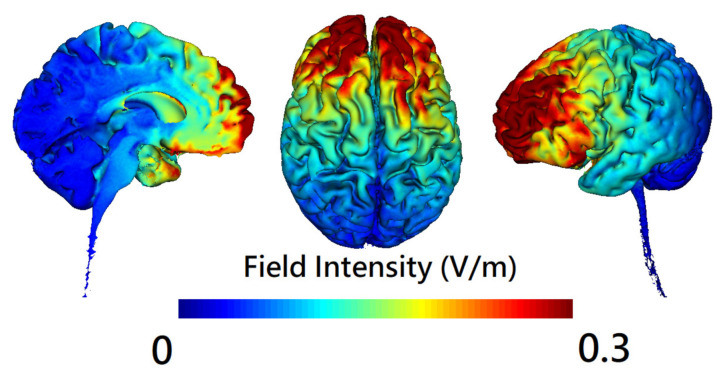
The modeling of electric field distribution of high-definition high-frequency transcranial random noise stimulation (hf-tRNS) over the lateral prefrontal cortex. The figure was created using HD-Explore^®^ software (Soterix Medical, New York, NY, USA).

**Figure 3 jpm-12-01617-f003:**
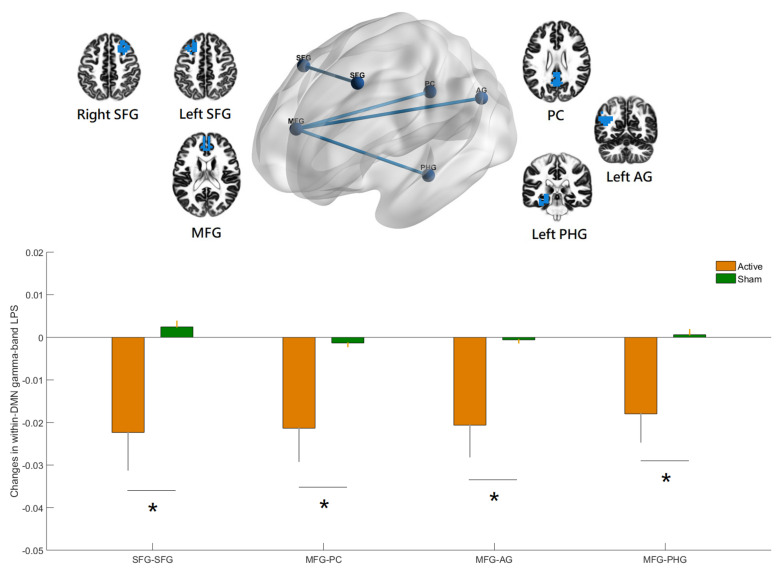
Relative to the sham, the hf-tRNS group had significant decreases in gamma-band within-network lagged phase synchronization (LPS) in the default mode network (DMN) from baseline to the end of stimulation (all *p* values < 0.05, corrected), specifically between the medial frontal gyrus (MFG) and the left angular gyrus (AG), between the MFG and the left parahippocampal gyrus (PHG), between the MFG and the posterior cingulate (PC), and between the right and the left superior frontal gyrus (SFG). The figure was created using eLORETA and BrainNet Viewer. Regions of interest (ROIs) shown here are displayed on a 5 × 5 × 5 MNI template brain in eLORETA for analyses (5 mm resolution is used). Error bars indicated standard errors. * *p* < 0.05 (corrected).

**Figure 4 jpm-12-01617-f004:**
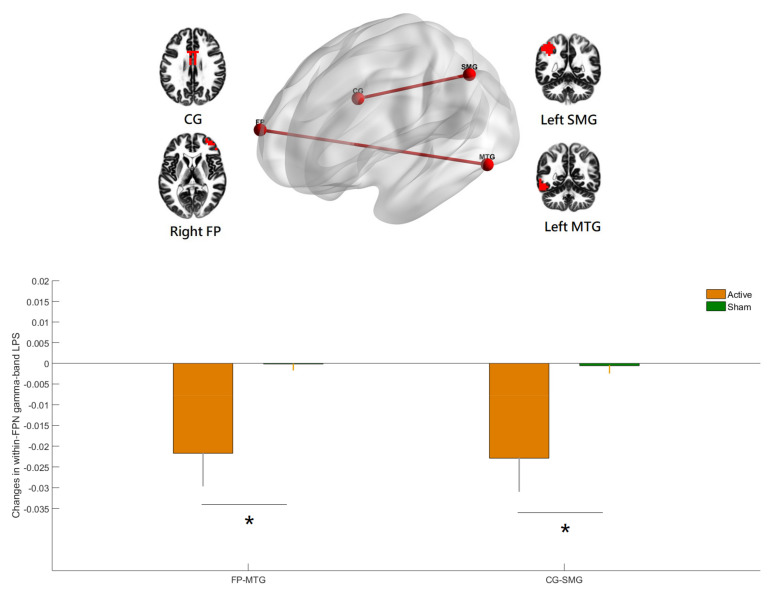
Relative to the sham, the hf-tRNS group had significant decreases in gamma-band within-network lagged phase synchronization (LPS) in the fronto-parietal network (FPN) from baseline to the end of stimulation (all *p* values < 0.05, corrected)**,** specifically, between the right frontal pole (FP) and the left middle temporal gyrus (MTG) and between the left supramarginal gyrus (SMG) and the cingulate gyrus (CG). The figure was created using eLORETA and BrainNet Viewer. Regions of interest (ROIs) shown here are displayed on a 5 × 5 × 5 MNI template brain in eLORETA for analyses (5 mm resolution is used). Error bars indicated standard errors. * *p* < 0.05 (corrected).

**Table 1 jpm-12-01617-t001:** Patient demographics and clinical data.

	hf-tRNS (N = 17)	Sham (N = 18)
Gender (f/m)	6/11	8/10
Handedness (r/l)	15/2	16/2
Age, years old	44.06 ± 12.50	43.17 ± 11.63
Years of education, years	13.53 ± 2.32	12.44 ± 3.52
Years since diagnosis, years	18.82 ± 9.73	19.11 ± 13.35
Chlorpromazine equivalent dose, mg/day	581.70 ± 310.59	626.10 ± 298.82
PANSS total score	69.00 ± 9.64	71.39 ± 9.06
PANSS Factor Score for Negative Symptoms (FSNS)	21.29 ± 3.00	21.94 ± 4.40
PANSS Factor Score for Positive Symptoms	11.71 ± 3.46	12.61 ± 3.74
PANSS Factor Score for Excitement	5.59 ± 2.48	5.28 ± 1.81
PANSS Factor Score for Disorganization	11.88 ± 1.27	12.11 ± 2.14
PANSS Factor Score for Emotional distress	5.82 ± 2.46	6.39 ± 1.50

Abbreviations: hf-tRNS, High-frequency transcranial random noise stimulation; PANSS,.Positive and Negative Syndrome Scale; FSNS, Factor Score for Negative Symptoms. Notes: Data are presented as means ± standard deviations unless otherwise stated.

**Table 2 jpm-12-01617-t002:** MNI coordinates for the seeds from the DMN and FPN.

Network	Anatomic Structure	Side	XYZ (MNI)		
DMN	Superior frontal gyrus	L	−27	23	48
	Superior frontal gyrus	R	27	23	48
	Angular gyrus	L	−41	−60	29
	Angular gyrus	R	41	−60	29
	Middle temporal gyrus	L	−64	−20	−9
	Middle temporal gyrus	R	64	−20	−9
	Medial frontal gyrus	Mid	0	49	18
	Parahippocampal gyrus	L	−25	−32	−18
	Parahippocampal gyrus	R	25	−32	−18
	Posterior cingulate	Mid	0	−52	26
FPN	Frontal pole	L	−40	50	7
	Frontal pole	R	40	50	7
	Supramarginal gyrus	L	−43	−50	46
	Supramarginal gyrus	R	43	−50	46
	Middle temporal gyrus	L	−57	−54	−9
	Middle temporal gyrus	R	57	−54	−9
	Paracingulate gyrus	Mid	0	22	47
	Cingulate gyrus	Mid	0	4	29
	Precuneus cortex	Mid	0	−76	45

Abbreviations: MNI, Montreal Neurological Institute; DMN, Default Mode Network; FPN, Fronto-Parietal Network. Notes: Coordinates are in MNI space. L = Left hemisphere seed; R = Right hemisphere seed; Mid = Midline seed. X = left (−) to right (+); Y = posterior (−) to anterior (+); Z = inferior (−) to superior (+).

## Data Availability

The data presented in this study are available on request from the corresponding author.

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
