# Peer review of "High-Frequency Transcranial Random Noise Stimulation Modulates Gamma-Band EEG Source-Based Large-Scale Functional Network Connectivity in Patients with Schizophrenia: A Randomized, Double-Blind, Sham-Controlled Clinical Trial"

_jpm, 2022, doi:10.3390/jpm12101617_

Round 1

Reviewer 1 Report

Reviewers' Comments to the Authors:

This in an interesting study that authors evaluated the effects of high-frequency transcranial random noise stimulation on gamma-band EEG source-based large-scale functional network connectivity in patients with schizophrenia in a randomized, double-blind, sham-controlled clinical trial.  There are however some issues that need to be addressed before this manuscript can be considered for publication (see my Comments below).

SPECIFIC COMMENTS TO THE AUTHORS:

1. Title:

-Title can be changes as followed:

High-frequency transcranial random noise stimulation modulates gamma-band EEG source-based large-scale functional network connectivity in patients with schizophrenia: a randomized, double-blind, sham-controlled clinical trial

2- Abstract

-The number of cases in each group should be mentioned. The sham group should be defined briefly.

-The keywords should be capitalized and selected using MESH.

3- Introduction

To be logic, authors should first explain the problem and then focus on the treatment methods and superiority of High-frequency transcranial random noise stimulation compared to other methods should be explained.  

5- Methods:

-A CONSORT Flow Diagram can be design for this study.  

-Was it a blinded study? Authors can explain who was blind for the procedure.

-How sample size was calculated? How the randomization was performed? Was there any special technique?

-I think author can use a CONSORT check list and upload it as a supplementary material of their study.

-Was there any study to evaluate the improvement of negative symptoms following investigation?

6-Results

-The values in each part can be shown in appropriated graphs to better show the results and comparisons. Also, correlations can be shown in a table or designed graphs. 

7-Discussion

- The Discussion section should be totally rewritten. the results should be compared and explained. 

- Limitations of study can be put under a subtitle.  

GENERAL COMMENTS TO THE AUTHORS:

1. Language: Some sentences are awkwardly phrased and should be re-written to improve the readability of the manuscript

Author Response

Authors’ response to the comments from Reviewer 1  

This in an interesting study that authors evaluated the effects of high-frequency transcranial random noise stimulation on gamma-band EEG source-based large-scale functional network connectivity in patients with schizophrenia in a randomized, double-blind, sham-controlled clinical trial.  There are however some issues that need to be addressed before this manuscript can be considered for publication (see my Comments below).

SPECIFIC COMMENTS TO THE AUTHORS:

  1. Title:

-Title can be changes as followed:

High-frequency transcranial random noise stimulation modulates gamma-band EEG source-based large-scale functional network connectivity in patients with schizophrenia: a randomized, double-blind, sham-controlled clinical trial

  • Response: Thanks for the suggestion. We revise the title.

2- Abstract

-The number of cases in each group should be mentioned. The sham group should be defined briefly.

-The keywords should be capitalized and selected using MESH.

  • Response: Thanks for the suggestion. We revise them accordingly.

3- Introduction

To be logic, authors should first explain the problem and then focus on the treatment methods and superiority of High-frequency transcranial random noise stimulation compared to other methods should be explained.  

  • Response: Thanks for the precise suggestion. We explain them in the introduction accordingly.

5- Methods:

-A CONSORT Flow Diagram can be design for this study.  

-Was it a blinded study? Authors can explain who was blind for the procedure.

-How sample size was calculated? How the randomization was performed? Was there any special technique?

-I think author can use a CONSORT check list and upload it as a supplementary material of their study.

-Was there any study to evaluate the improvement of negative symptoms following investigation?

  • Response: Thanks for the suggestion. We revise them in the supplementary materials accordingly.

6-Results

-The values in each part can be shown in appropriated graphs to better show the results and comparisons. Also, correlations can be shown in a table or designed graphs. 

  • Response: Thanks for the suggestion. The values, graphs and the correlation table are added

7-Discussion

- The Discussion section should be totally rewritten. the results should be compared and explained. 

- Limitations of study can be put under a subtitle.  

  • Response: Thanks for the valuable suggestion. We revise them accordingly.

GENERAL COMMENTS TO THE AUTHORS:

  1. Language: Some sentences are awkwardly phrased and should be re-written to improve the readability of the manuscript

  • Response: Thanks for the valuable suggestion. We refine the manuscript accordingly.

Reviewer 2 Report

Hi Dears

Thanks for this research and Novelty ideas.

Please explain more about Diagnosis , sampling and Exclude criteria.

Author Response

Authors’ response to the comments from Reviewer 2

Hi Dears

Thanks for this research and Novelty ideas.

Please explain more about Diagnosis , sampling and Exclude criteria.

  • Response: Thanks for the valuable suggestion. We revise them accordingly.

Reviewer 3 Report

I have read with great interest the study entitled “High-frequency transcranial random noise stimulation modulates gamma-band EEG source-based large-scale functional network connectivity in patients with schizophrenia”. In this manuscript, the authors report a TMS experiment on schizophrenic patients.

Introduction, Lines 70-77 & Discussion. The authors did a good work in summarizing studies investigating FPN and DMN involvement in schizophrenia. However, many studies also shown the involvement of these brain networks in healthy individuals with high psychotic traits (Hua et al., 2019; Di Plinio et al., 2020, Koban et al., 2021). The authors should discuss the interconnectedness of their findings with studies on healthy individuals to furnish a more complete overview of brain basis of schizophrenia.

Methods, Section 2.1. It is appropriate that the authors refer to their other paper. However, the manuscript should be exhaustive also without mandatorily having to search and obtain other papers. Thus, I invite the authors to summarize critical information such as exclusion/inclusion criteria, and generally all of this information that are necessary to make the present manuscript complete.

Methods, Section 2.2, Line 120. What does "equally" mean here? I think it is not necessary.

Methods, Section 2.2. Were the two groups matched for symptoms? In other words, was the randomization of sham vs treatment controlled in some way? Please report scores (average, standard deviation) for each group.

Methods, Section 2.4, Lines 169-170. This is a good procedure. How did the authors use such information to pre-process EEG data? Please describe.

Methods, Section 2.4, Line 179. Please briefly describe FIR filtering.

Methods, Section 2.6. Seed-based analyses with a-priori selected seeds are substandard and biased analyses, are discouraged by modern guidelines in neurosciences, should not be performed, and cannot be accepted alone in peer-reviewed publications. Please report whole-brain analyses to avoid biases.

Discussion. The authors state that “Previous research on target engagement of gamma 296 oscillations by hf-tRNS has been limited by healthy control participants […]”. However, their study lack of a control group of non-schizophrenic individuals. This is a limitation of the study that must be reported and discussed.

REFERENCES

Di Plinio, S, Perrucci, MG, Aleman, A, Ebisch, SJH (2020). I am Me: Brain systems integrate and segregate to establish a multidimensional sense of self. NeuroImage, 205:116284. doi: 10.1016/j.neuroimage.2019.116284.

Hua, JPY, Karcher, NR, Merrill, AM, O’Brien, KJ, Straub, KT, Trull, TJ, Kerns, JG (2019). Psychosis risk is associated with decreased resting-state functional connectivity between the striatum and the default mode network. Cogn Affect Behav Neurosci, 19(4):998-1011. doi: 10.3758/s13415-019-00698-z.

Koban, L, Gianaros, PJ, Kober, H, Wager, TD (2021). The self in context: brain systems linking mental and physical health. Nat Rev Neurosci, 22(5):309-322. doi: 10.1038/s41583-021-00446-8.

Author Response

Authors’ response to the comments from Reviewer 3

I have read with great interest the study entitled “High-frequency transcranial random noise stimulation modulates gamma-band EEG source-based large-scale functional network connectivity in patients with schizophrenia”. In this manuscript, the authors report a TMS experiment on schizophrenic patients.

Introduction, Lines 70-77 & Discussion. The authors did a good work in summarizing studies investigating FPN and DMN involvement in schizophrenia. However, many studies also shown the involvement of these brain networks in healthy individuals with high psychotic traits (Hua et al., 2019; Di Plinio et al., 2020, Koban et al., 2021). The authors should discuss the interconnectedness of their findings with studies on healthy individuals to furnish a more complete overview of brain basis of schizophrenia.

  • Response: Thanks for the valuable suggestion. We revise them accordingly.

Methods, Section 2.1. It is appropriate that the authors refer to their other paper. However, the manuscript should be exhaustive also without mandatorily having to search and obtain other papers. Thus, I invite the authors to summarize critical information such as exclusion/inclusion criteria, and generally all of this information that are necessary to make the present manuscript complete.

  • Response: Thanks for the valuable suggestion. We revise them accordingly.

Methods, Section 2.2, Line 120. What does "equally" mean here? I think it is not necessary.

  • Response: Thanks for the valuable suggestion. We delete it.

Methods, Section 2.2. Were the two groups matched for symptoms? In other words, was the randomization of sham vs treatment controlled in some way? Please report scores (average, standard deviation) for each group.

  • Response: Thanks for the valuable suggestion. We revise them accordingly.

Methods, Section 2.4, Lines 169-170. This is a good procedure. How did the authors use such information to pre-process EEG data? Please describe.

  • Methods, Section 2.4, Line 179. Please briefly describe FIR filtering.

 Response: Thanks for the valuable suggestion. We briefly describe FIR filtering accordingly.

Methods, Section 2.6. Seed-based analyses with a-priori selected seeds are substandard and biased analyses, are discouraged by modern guidelines in neurosciences, should not be performed, and cannot be accepted alone in peer-reviewed publications. Please report whole-brain analyses to avoid biases.

  • Response: Thanks for the valuable suggestion. We report it

Discussion. The authors state that “Previous research on target engagement of gamma 296 oscillations by hf-tRNS has been limited by healthy control participants […]”. However, their study lack of a control group of non-schizophrenic individuals. This is a limitation of the study that must be reported and discussed.

  • Response: Thanks for the valuable suggestion. We revise them accordingly. .

REFERENCES

Di Plinio, S, Perrucci, MG, Aleman, A, Ebisch, SJH (2020). I am Me: Brain systems integrate and segregate to establish a multidimensional sense of self. NeuroImage, 205:116284. doi: 10.1016/j.neuroimage.2019.116284.

Hua, JPY, Karcher, NR, Merrill, AM, O’Brien, KJ, Straub, KT, Trull, TJ, Kerns, JG (2019). Psychosis risk is associated with decreased resting-state functional connectivity between the striatum and the default mode network. Cogn Affect Behav Neurosci, 19(4):998-1011. doi: 10.3758/s13415-019-00698-z.

Koban, L, Gianaros, PJ, Kober, H, Wager, TD (2021). The self in context: brain systems linking mental and physical health. Nat Rev Neurosci, 22(5):309-322. doi: 10.1038/s41583-021-00446-8.

Round 2

Reviewer 1 Report

The randomization and blinding and sample size calculation can be transferred from supplementary material to the main text, article can be accepted after this minor revision. 

Author Response

Response: Thanks for the suggestion. We transfer the randomization and blinding and sample size calculation from supplementary material to the main text.

Reviewer 3 Report

I thank the authors for revising their manuscript. The revision was appropriate and I praise tha authors for that. The manuscript is way more clear and comprehensive now.

However, I have two statements.

First, I find the way the authors replied to reviewers' comments inappropriate and almost insulting. I spent time and energy reviewing their manuscript, but they just answered using frigid short sentences which do not illustrate their revision process nor their scientific stream of thought. Usually, I would reject a manuscript with this type of superficial and pointless answers. I kindly suggest the author to spend more time in communicating with reviewers, for their next publications. Communication during revision is central in the societal scientific process, and in this manuscript they are volountarily reducing their scientific potential.

Second, please note that whole-brain results are more relevant and more trustable that results after cherry-picking ROIs. Report such results in the main text and integrate them more significantly in the manuscript.

Author Response

Authors’ response to the comments from Reviewer 3

I thank the authors for revising their manuscript. The revision was appropriate and I praise tha authors for that. The manuscript is way more clear and comprehensive now.

However, I have two statements.

Comment 1

First, I find the way the authors replied to reviewers' comments inappropriate and almost insulting. I spent time and energy reviewing their manuscript, but they just answered using frigid short sentences which do not illustrate their revision process nor their scientific stream of thought. Usually, I would reject a manuscript with this type of superficial and pointless answers. I kindly suggest the author to spend more time in communicating with reviewers, for their next publications. Communication during revision is central in the societal scientific process, and in this manuscript they are volountarily reducing their scientific potential.

Response 1

We are very sorry about our unfamiliarity of correctly uploading to the submission system the pdf file “Authors’ response to the comments from Reviewer 3 (Complete version of round 1 review)” as can be seen in the following supplementary texts. Therefore, the reviewer 3 can only read “the concise version of authors’ response to the comments from Reviewer 3” that is not a point-by-point response to the reviewer’s comments.   

Comment 2

Second, please note that whole-brain results are more relevant and more trustable that results after cherry-picking ROIs. Report such results in the main text and integrate them more significantly in the manuscript.

Response 2

We thank the reviewer’s insightful comment. We add figure S3 and Table S4, and also revise Table S3. We revise the results on page 11, line 376-383 “Figures S2-3 and Table S3 showed that patients treated with hf-tRNS had reduced functional connectivity in several large-scale brain networks at the end of stimulation in comparison with the sham group (e.g., connectivity between right middle frontal gyrus and left posterior cingulate, between right middle frontal gyrus and left cuneus, between right lingual gyrus and bilateral anterior cingulate, between right cingulate gyrus and right paracentral lobule, and between right superior temporal gyrus and left posterior cingulate, all t values > 3.22, two-tailed p values < 0.01, Table S4).” We also revise the discussion on page 13-14, line 480-485 “It is noteworthy that hf-tRNS reduced functional connectivity of cuneus, lingual gyrus, superior temporal gyrus and paracentral lobule in large-scale brain networks, as can be seen in the results of whole-brain analyses (Figure S3 and Table S4). Research indicates that these networks play an unneglectable role in the pathophysiology of schizophrenia [36, 37]. ” and on page 14, line 488-492 “The limitation can be overcome by future studies applying high-density (256 channel) scalp EEG recording system that has significantly improved spatial resolution for source localization of EEG signals on the cortical surface and has competitive reliability and agreement of power envelope connectivity [38, 39].”  In addition, new references 36-39 are added as well.

Supplementary texts

-----------------------------------------------------------------------------------------------------------------

Authors’ response to the comments from Reviewer 3 (Complete version of round 1 review)

I have read with great interest the study entitled “High-frequency transcranial random noise stimulation modulates gamma-band EEG source-based large-scale functional network connectivity in patients with schizophrenia”. In this manuscript, the authors report a TMS experiment on schizophrenic patients.

Comment 1

 Introduction, Lines 70-77 & Discussion. The authors did a good work in summarizing studies investigating FPN and DMN involvement in schizophrenia. However, many studies also shown the involvement of these brain networks in healthy individuals with high psychotic traits (Hua et al., 2019; Di Plinio et al., 2020, Koban et al., 2021). The authors should discuss the interconnectedness of their findings with studies on healthy individuals to furnish a more complete overview of brain basis of schizophrenia.

Response 1

Thanks for the valuable suggestion. We revise the introduction on page 2, Line 81-89.

Increasing evidence suggests that functional integration and segregation these large-scale neuronal networks are critical for self-experience and mental health [10]. Specifically, aberrant functional interactions within these brain networks may have particular relevance for self-experience alterations (e.g., psychosis-like experiences) and an increased risk for  psychosis in healthy individuals [11, 12]. Neuroimaging studies in schizophrenia have  depicted that the aberrant connections within and between DMN and FPN as revealed by resting-state functional magnetic resonance imaging (fMRI) are correlated with negative symptom burden and negative symptom reductions following treatment [9, 13].

Comment 2

Methods, Section 2.1. It is appropriate that the authors refer to their other paper. However, the manuscript should be exhaustive also without mandatorily having to search and obtain other papers. Thus, I invite the authors to summarize critical information such as exclusion/inclusion criteria, and generally all of this information that are necessary to make the present manuscript complete.

Response 2

Thanks for the valuable suggestion. We revise the part of methods on page 3, line 126-144 and add supplementary material to further make the present manuscript complete.

Comment 3

Methods, Section 2.2, Line 120. What does "equally" mean here? I think it is not necessary.

Response 3

Thanks for the valuable suggestion. We delete it on page 4, line 153-154  

Comment 4

Methods, Section 2.2. Were the two groups matched for symptoms? In other words, was the randomization of sham vs treatment controlled in some way? Please report scores (average, standard deviation) for each group.

Response 4

Thanks for the valuable suggestion. We add supplementary material to provide detailed information of randomization. We did not control negative symptom severity between active and sham group during the procedure of randomization. However, we add a description on Page 3, line 142-145 “There were no significant differences in the sociodemographic and clinical characteristics between hf-tRNS group and sham group.” for a clear description of the baseline between-group different in negative symptom severity.

Comment 5

Methods, Section 2.4, Lines 169-170. This is a good procedure. How did the authors use such information to pre-process EEG data? Please describe.

Response 5

Thanks for the valuable suggestion. On page 6, line 216-217, we provide supplementary material to provide detailed information of the pre-processing of EEG data for online correction of eye movement and artifacts.

Comment 6

Methods, Section 2.4, Line 179. Please briefly describe FIR filtering.

 Response 6

Thanks for the valuable suggestion. We briefly describe FIR filtering by revising the method on page 6, line 223-225 “Offline, the data were downsampled to 500 Hz, 222 band-pass filtered to 1–100 Hz with the Finite Impulse Response (FIR) method [i.e., 223 using the tool of Basic FIR filter (new, default) with 1 Hz as the lower edge frequency 224 and 100 Hz as the higher edge frequency]”.  

Comment 7

Methods, Section 2.6. Seed-based analyses with a-priori selected seeds are substandard and biased analyses, are discouraged by modern guidelines in neurosciences, should not be performed, and cannot be accepted alone in peer-reviewed publications. Please report whole-brain analyses to avoid biases.

Response 7

 Thanks for the valuable suggestion. We report it in the figure S2 of supplementary material and revise the results on page 11-12, line 346-351.

 3.4. Effects of hf-tRNS on source-based gamma-band whole-brain functional connectivity

In addition to eLORETA seed-based analyses with a-priori selected seeds for large-scale network functional connectivity analyses, the whole-brain analyses were further reported in the supplementary materials to avoid biases. Figure S2 showed that patients treated with hf-tRNS had reduced functional connectivity in several large-scale 350 brain networks at the end of stimulation in comparison with the sham group.

Comment 8

Discussion. The authors state that “Previous research on target engagement of gamma oscillations by hf-tRNS has been limited by healthy control participants […]”. However, their study lack of a control group of non-schizophrenic individuals. This is a limitation of the study that must be reported and discussed.

Response 8

Thanks for the valuable suggestion. We revise the discussion on page 14, line 493-498 “Fourth, the statement and interpretation regarding the resting-state gamma activity or gamma-band large-scale network connectivity in our patients with schizophrenia should be treated with caution because this study did not have a control group of non-schizophrenia individuals. Further studies including both patients and healthy subjects are required to confirm our results.”

REFERENCES

Di Plinio, S, Perrucci, MG, Aleman, A, Ebisch, SJH (2020). I am Me: Brain systems integrate and segregate to establish a multidimensional sense of self. NeuroImage, 205:116284. doi: 10.1016/j.neuroimage.2019.116284.

Hua, JPY, Karcher, NR, Merrill, AM, O’Brien, KJ, Straub, KT, Trull, TJ, Kerns, JG (2019). Psychosis risk is associated with decreased resting-state functional connectivity between the striatum and the default mode network. Cogn Affect Behav Neurosci, 19(4):998-1011. doi: 10.3758/s13415-019-00698-z.

Koban, L, Gianaros, PJ, Kober, H, Wager, TD (2021). The self in context: brain systems linking mental and physical health. Nat Rev Neurosci, 22(5):309-322. doi: 10.1038/s41583-021-00446-8.

Round 3

Reviewer 3 Report

I thank the authors for revising their manuscript. The revision was appropriate and I praise tha authors for that. The manuscript is way more clear and comprehensive now.